# Novel Organic Material Induced by Electron Beam Irradiation for Medical Application

**DOI:** 10.3390/polym12020306

**Published:** 2020-02-03

**Authors:** Adrian Barylski, Krzysztof Aniołek, Andrzej S. Swinarew, Sławomir Kaptacz, Jadwiga Gabor, Zbigniew Waśkiewicz, Arkadiusz Stanula

**Affiliations:** 1Institute of Materials Engineering, Faculty of Science and Technology, University of Silesia in Katowice, 75 Pułku Piechoty 1A, 41-500 Chorzów, Poland; adrian.barylski@us.edu.pl (A.B.); krzysztof.aniolek@us.edu.pl (K.A.); andrzej.swinarew@us.edu.pl (A.S.S.); slawomir.kaptacz@us.edu.pl (S.K.); jadwiga.gabor@us.edu.pl (J.G.); 2Institute of Sport Science, The Jerzy Kukuczka Academy of Physical Education, Mikołowska 72A, 40-065 Katowice, Poland; z.waskiewicz@awf.katowice.pl

**Keywords:** PTFE, electron beam irradiation, thermal properties, DSC, FTIR

## Abstract

This study analyzed the effects of irradiation of polytetrafluoroethylene (PTFE) containing 40% of bronze using an electron beam with energy of 10 MeV. Dosages from 26 to156 kGy (2.6–15.6 Mrad) were used. The impact of a high-energy electron beam on the thermal, spectrophotometric, mechanical, and tribological properties was determined, and the results were compared with those obtained for pure PTFE. Thermal properties studies showed that such irradiation caused changes in melting temperature *T_m_* and crystallization temperature *T_c_*, an increase in crystallization heat *∆H_c_*, and a large increase in crystallinity *χ*_c_ proportional to the absorbed dose for both polymers. The addition of bronze decreased the degree of crystallinity of PTFE by twofold. Infrared spectroscopy (FTIR) studies confirmed that the main phenomenon associated with electron beam irradiation was the photodegradation of the polymer chains for both PTFE containing bronze and pure PTFE. This had a direct effect on the increase in the degree of crystallinity observed in DSC studies. The use of a bronze additive could lead to energy dissipation over the additive particles. An increase in hardness H and Young’s modulus E was also observed. The addition of bronze and the irradiation with an electron beam improved of the operational properties of PTFE.

## 1. Introduction

Polytetrafluoroethylene (PTFE) was first obtained by Roy Plunkett in 1938 by accident during the synthesis of new refrigerants [1]. The advances in the development of stable PTFE production processes was influenced by the outbreak of World War II, and the unusual combination of properties of this polymer led to its use as a sealing material in the construction of the atomic bomb in the Manhattan project [2].

The properties of PTFE result from its structure consisting in a twisted helix (13 CF_2_ groups every 180° of rotation) (Figure 1). The helix forms an almost ideal cylinder in which fluorine atoms surround the carbon core [3]. The structure of PTFE resembles parallel rods that can slide relatively to each other. The mutual repulsion of the fluorine atoms tends to inhibit the bending of the polymer backbone chain, resulting in a large increase in stiffness. In turn, the high chemical inertness and stability of PTFE, as well as its low coefficient of friction and stickiness, are caused by the outer cover of fluorine atoms which protects the carbon skeleton [4].

The exceptional ultraviolet (UV) radiation, chemical resistance, and thermal stability can be explained by the fact that the C–F and C–C bonds in fluorinated derivatives are among the strongest bonds in organic compounds [1]. PTFE is widely used in different branches of industry (electronics, mechanical engineering, aerospace, chemical industries, and medical engineering) because it has good thermal properties, high chemical and biological resistance, and good dielectric and tribological properties (antifriction, antiadhesive) [5,6,7]. PTFE is also one of the best polymers available on the market for tribological applications. Due to its structure (extended-chain linear molecules, –(CF_2_–CF_2_)_n_–), it can transfer a thin sliding film onto surfaces interacting during friction, significantly reducing the friction coefficient [8,9,10,11]. However, because contaminants in the form of lamellae, nodules, and slabs of polymer, with dimensions of several tenths of a millimeter, are also transferred during the tribological interaction of pure PTFE, PTFE without any additives is characterized by a very high wear rate [8].

The application of various types of additives (graphite, bronze, molybdenum disulfide) results in the reduction of the wear values by nearly three times. The use of bronze as an additive also causes an increase in heat resistance (operating temperature from −260 °C to +260 °C), better sliding and friction properties compared to those of pure PTFE, increased stiffness and hardness of the final material, better materials for wicking, physiological inertness, and low expansion coefficient. Another way to improve the properties of PTFE is to apply irradiation with an electron or gamma beam during the interaction with a smooth counter partner [12,13,14]. The majority of current research focuses on irradiation of pure PTFE with a beam of gamma rays, due to the limited availability of sources of electron irradiation with enough energy (in the order of MeV) to penetrate PTFE to a depth of several centimeters [15].

The purpose of this study was to determine the effect of 10 MeV electron beam irradiation on the properties of PTFE containing 40% bronze and to compare the results obtained with those for pure PTFE. The possibility of modifying the properties of PTFE by using only the electron beam at room temperature would significantly reduce the costs associated with the modifications and contribute to the improvement of the operating characteristics.

## 2. Materials and Methods

The research material was PTFE with 40% bronze content. As an additive, tin bronze without lead (10% Sn, 2% Zn, Pb < 0.004%) and with a grain size below 32 µm was used, and its properties were compared with those of pure PTFE (Inbras, Tarnów, Poland). The PTFE–bronze composite was made by cold pressing and sintering, and the pure PTFE was extruded. Electron beam irradiation was performed at the Institute of Nuclear Chemistry and Technology in Warsaw using the Elektronika 10/10 accelerator (10 MeV beam energy, 10 kW beam power, NPO Torij, Moscow, Russia). The absorbed dose for both tested materials was 26 kGy (2.6 Mrad), up to the dose of 156 kGy (15.6 Mrad). Irradiation was carried out at room temperature (21 ± 1 °C) in a vacuum.

### 2.1. Examination of the Polymers Thermal Properties

Differential scanning calorimetry (DSC) was used to assess the thermal properties. Sections weighting 15 mg were cut out from the central part of the specimens in the initial state and after irradiation with an electron beam. The examination was performed using a dynamic differential calorimeter, Mettler-Toledo DSC 1 (Mettler-Toledo GmbH, Greifensee, Switzerland), closing the specimens in aluminum cells. The rate of heating was 10 °C/min, and the temperature ranged from −40 °C to 400 °C. The samples in the molten state were recrystallized by cooling at the same rate (10 °C/min). Next, the thermograms were analyzed, and calculations of the crystallinity degree, *χ_c_*, were made according to the formula [16]:(1)$$\chi_{C}=\frac{\Delta H_{C}}{\Delta H_{f}}\cdot 100\left(\%\right)$$
where Δ*H_c_* is the heat of phase transition (i.e., cooling) of the investigated polymer sample, determined from a DSC thermogram (J/g); Δ*H_f_* is the heat of phase transition of completely crystalline PFTE (empirically determined value amounting to 82 J/g). In a previous study [17], it was shown that the heat of crystallization (phase transition) Δ*H_c_* of the investigated polymer samples is closely related to its molecular weight and can be used to calculate the number average molecular weight, *M_n_*, for PTFE:(2)$$M_{n}=\;2.1\times 10^{10}\cdot \Delta H_{c}^{-5.16}$$

### 2.2. FTIR Measurement of the Polymers

Spectrometric studies were carried out using an FTIR/ATR Shimadzu IR Tracer 100 Fourier Spectrometer (Shimadzu, Kyoto, Japan). The FTIR spectra were obtained in the spectral region of 4000–550 cm^−1^. To determine the functional groups of the samples, the attenuated total reflection–FTIR (ATR–FTIR) technique was used. This is one of the most accurate spectroscopic methods which allows assigning particular functional groups to specific areas where characteristic absorption bands are present. Reflectance was used for surface analysis of soft samples and liquids. The infrared (IR) beam passed a few microns into the sample before it was reflected. Contact between the crystal and the sample was achieved. All measurements were taken at ambient temperature (21 ± 1 °C).

### 2.3. Examination of The Polymers’ Mechanical Properties

A Micron-Gamma tester (manufactured by the Faculty of Aviation and Space Systems: The National Technical University of Ukraine “Igor Sikorsky Kyiv Polytechnic Institute”) equipped with a self-leveling table was used for micromechanical properties tests. The Berkovich penetrator was used in microindentation tests, with the maximum load of *P* = 1 N, time of loading and unloading of −30 s, and time of holding under the maximum load of −10 s. Measurements were made following the ISO 14577 standard. To determine hardness H and Young’s modulus E, the standard Oliver–Pharr method was used [18]. Measurements were made at room temperature, and results were averages from 10 indents.

### 2.4. Examination of the Polymers Tribological Properties

The tribological tests of PTFE containing 40% bronze and without additives were performed using a pin-on-disk tribometer T-01 (manufactured by IteE, Radom, Poland). Three samples in the form of pins (5 mm of diameter) were prepared for both starting materials as well as for materials exposed to irradiation at varying degrees. As a counter partner, discs made of 1H18N9T steel with surface roughness Ra = 0.2 µm were used. Such coupling of pin and disk enabled the formation of a thin film limiting the coefficient of friction and wear. Tests were performed in accordance with the recommendations of VAMAS Technical Note and the requirements of the ASTM G-99 standard [19,20], with the following contact parameters: dry friction, pin diameter of 5 mm, friction distance diameter of 24 mm, sliding speed of 0.1 m/s, load of 40 N (pressure: 2 MPa), and friction distance of 1000 m. Ambient parameters were: temperature = 21 ± 1 °C and humidity = 50 ± 5%. The linear wear *W*_L_ was determined as the difference between the indications of the micrometric sensor before and after the test (and after the cooling stage).

## 3. Results and Discussion

### 3.1. Thermal Studies

DSC studies confirmed that the interaction with a 10 MeV electron beam had a great impact on the thermal properties of pure PTFE as well as of a PTFE composite with 40% bronze. The changes in the thermal properties of the polymers in their initial states and after irradiation are presented in Table 1 and Table 2. Parameters such as melting point *T*_m_ (during heating), crystallization temperature *T*_c_ (during cooling), melting heat Δ*H*_m_, heat of crystallization Δ*H**_c_***, degree of crystallinity *χ*_c_, and average molecular weight *M*_n_ are specified.

By analyzing the data from Table 1 and Table 2 and Figure 2 (showing DSC thermograms from the heating process for pure PTFE and for PTFE with 40% bronze), it was noticed that the intensity of the electron beam interactions was greater for the pure material than for the composite. For pure PTFE, an increase in melting point *T*_m_ and a significant increase in melting heat Δ*H*_m_ were already observed at an absorbed dose of 26 kGy. A further increase in the irradiation dose corresponding to 52–156 kGy caused small changes in temperature and melting heat.

The formation of an additional melting peak around 325 °C was also visible for absorbed doses from 78 to 156 kGy. This was due to the breaking of the PTFE chains and the fragmentation of the polymer crystal structure occurring during irradiation at room temperature. Similar results were published by Abdou and Mohamed [21]. By analyzing the results of the PTFE–bronze composite, it was noticed that, in its initial state, this composite had about a 60% lower melting heat compared to pure PTFE. However, the intensity of Δ*H*_m_ growth, caused by electron beam irradiation, remained at a similar level. There were no significant differences in the *T*_m_ of the composite at its initial state and after irradiation with an electron beam.

Analysis of DSC thermograms registered during the melt recrystallization process (Table 1 and Table 2 and Figure 3) allowed the determination of *T*_c_ and Δ*H*_c_.

By analyzing the results for both (a) PTFE and (b) PTFE–bronze composite, it could be observed that *T*_c_ was not show significantly influenced by the electron beam. Contrarily, a significant increase in Δ*H*_c_ and *χ_c_*, determined by the Formula (1), was observed (Figure 4) with an increase of the absorbed dose.

A lower *χ_c_* registered for the PTFE–bronze composite might indicate that, in this material, the phenomenon of electron beam scattering on additive particles occurred. Irradiation with an electron beam caused a gradual increase in *χ_c_* with similar intensity in both studied materials, and this increase could be explained by chain decomposition and the formation of additional crystallites in the amorphous areas of PTFE [21]. The Δ*H_c_* directly correlates with *M_n_* (Formula (2)) [17], which means that the higher the *χ_c_*, the lower the *M_n_* of the polymer (Figure 5). The shorter polymer chain size had a positive effect on rearrangement and orientation during the crystallization process due to greater mobility and less intermolecular entanglement.

PTFE with 40% bronze was characterized by a higher *M_n_* in the initial state and, therefore, a greater decrease in the *M_n_* values after electron beam irradiation.

### 3.2. FTIR Analysis

Structural vibration analysis and end-groups search indicated differences between PTFE samples exposed or unexposed to irradiation. The presence of bands in the range of 1000–1300 cm^−1^ and 525–800 cm^−1^ (asymmetric and symmetric stretching vibrations C–F) confirmed the presence of PTFE (Figure 6a). Electron beam irradiation of pure PTFE caused changes in the intensity of the FTIR spectrum, especially in the range of 525–800 cm^−1^ (Figure 6b).

The spectra had a greater intensity before sample irradiation, especially in comparison to spectra obtained after applying a 156 kGy irradiation dose. This phenomenon was caused by the different number of mer units in the main chain of the polymer. The number of mer units could be different due to the degradation process of PTFE under electron beam irradiation. All samples were exposed to a dose between 26 and 156 kGy.

In the case of the PTFE–bronze composite, the change in spectrum intensity was less visible (Figure 7); therefore, the doping process was highly correlated with changes in spectrum intensity. These results could be caused by energy dissipation on dopant particles with a grain size below 32 µm. Crystallinity increased due to a decrease in the length of the polymer chains. The irradiation of the samples resulted in the creation of a low-energy agglomerate, which was caused by the dissolution of the polymer backbone and presented statistically distributed crystalline regions. Changes related to the uprising crystalline phase and more crystalline agglomerates at low energy were observed for both materials. The obtained spectral changes were correlated with changes in the mechanical, tribological, and thermal properties.

### 3.3. Mechanical Properties of PTFE

The results of the thermal and spectroscopic studies confirmed that electron beam irradiation causes changes in the length of macromolecules in exposed polymers and, consequently, in their physical properties. Irradiation led to changes in the non-crystalline and crystalline content, and the degradation and scissions occurred mainly in the amorphous region. This suggested mechanism increased the degree of crystallinity, which had a direct impact on the mechanical properties of PTFE subjected to electron beam irradiation.

Base on the test results, electron beam irradiation of both tested materials caused an increase in hardness H (Figure 8a) and Young’s modulus E (Figure 8b), which was proportional to the absorbed radiation dose. Also, for materials in the initial state, it could be observed that PTFE with 40% bronze was characterized by a much higher hardness (by approx. 35%) and Young’s modulus (by approx. 43%) when compared to pure PTFE. Nevertheless, the improvement of the mechanical properties caused by electron beam irradiation was similar for the two polymers. The most favorable results were obtained for the samples that absorbed a 104 kGy dose of irradiation. Further irradiation promoted the degradation and deterioration of the mechanical properties, especially for pure PTFE.

### 3.4. Tribological Properties of PTFE

The increased degree of crystallinity and the changes in mechanical properties, induced by electron beam irradiation caused a significant improvement in the materials’ lifetime, reflected by their reduced tribological wear. Figure 9 shows the *W*_L_ of PTFE and PTFE with 40% bronze as a function of the absorbed radiation dose.

Initial PTFE was characterized by a very high *W*_L_, 2350.38 µm, which was significantly reduced as the absorbed dose of irradiation increased. PTFE after electron beam irradiation with a 104 kGy dose was characterized by having the most favorable tribological properties (over an 80-time reduction in linear wear, *W*_L_ = 28.88 µm, was observed). Further irradiation caused a slight increase in *W*_L_ to 53.88 µm. Figure 10 presents images of discs after tribological cooperation with pins made of PTFE, in the initial state and after electron beam irradiation. There was a significant reduction in the number of wear products and scratches of the friction pairs tested, proportionally to the dose of radiation used.

The use of bronze as an additive reduced tribological wear. PTFE with 40% bronze in its initial state was characterized by a *W*_L_ of 72.5 µm (over 30 times lower than that of pure PTFE). Electron beam irradiation of the polymer–bronze composite led to a reduction of wear by over 2.5 times. It is worth noting that the linear wear of pure PTFE irradiated with the 104 kGy dose and that of PTFE with 40% bronze before irradiation were at a similar level (approximately 28 µm). It can be concluded that irradiation with an electron beam at such a dose caused a similar reduction of linear wear as that induced by the additive. During the tribological tests, the friction coefficient was also measured in real time. The stabilized friction coefficient for pure PTFE was µ = 0.198 ± 0.01, and that for PTFE with 40% bronze was µ = 0.182 ± 0.01. The lower value of the friction coefficient for the composite was due to the fact that bronze additive during the tribological test acted as a solid grease, reducing the coefficient of friction. Irradiation with an electron beam did not affect the value of µ.

## 4. Conclusions

Irradiation with an electron beam caused changes in the thermal properties of PTFE and a PTFE–bronze composite. For the pure PTFE under the influence of irradiation, Δ*H*_m_ and *T*_m_ increased with the absorption of small doses (26 kGy). At higher doses (78–156 kGy), an additional melting peak was recorded on thermograms due to polymer chain decomposition and fragmentation of the crystalline structure. PTFE with 40% bronze was characterized by 60% lower melting heat compared to pure PTFE, maintaining the irradiation influence at a similar level. No significant changes in *T*_c_ of both tested materials were observed. However, a significant increase in Δ*H*_c_ and *χ*_c_ was observed. Due to the phenomenon of electron beam scattering on additive particles, a lower degree of crystallinity was observed in the PTFE–bronze composite.

An increase in the degree of crystallinity correlated with a decrease in the molecular weight of the polymer. In addition, shorter chains increased its mobility, allowing for easier rearrangement and orientation during the crystallization process. By analyzing the FTIR spectroscopic spectra, which showed changes of intensity primarily due to the decomposition of PTFE chains, we concluded that the use of the additive caused changes in the intensity of the bands due to energy dissipation on the dopant particles. The reconfiguration of the polymer structure by electron beam irradiation in both tested materials caused an increase in the hardness H and Young’s modulus E in proportion to the irradiation dose used. Furthermore, the use of a 40% bronze additive significantly increased the mechanical properties in the initial state. The most favorable results were obtained for samples irradiated with 104 kGy as, at higher doses, an increase in the degradation of the material was observed. The use of electron beam irradiation also reduced significantly the tribological wear. For the pure PTFE, there was a reduction of over 80 times in *W_L_* after absorbing a 104 kGy irradiation dose. The use of a 40% bronze additive reduced the wear by more than 30 times in the initial state. In the case of composite irradiation, a further reduction of 2.5 times in wear was observed, in proportion to the irradiation dose used. The modification of PTFE and PTFE bronze composites through irradiation with an electron beam might contribute to the extension of the life cycle of this material, e.g., in sliding components that work under heavy load conditions.

## Figures and Tables

**Figure 1 polymers-12-00306-f001:**
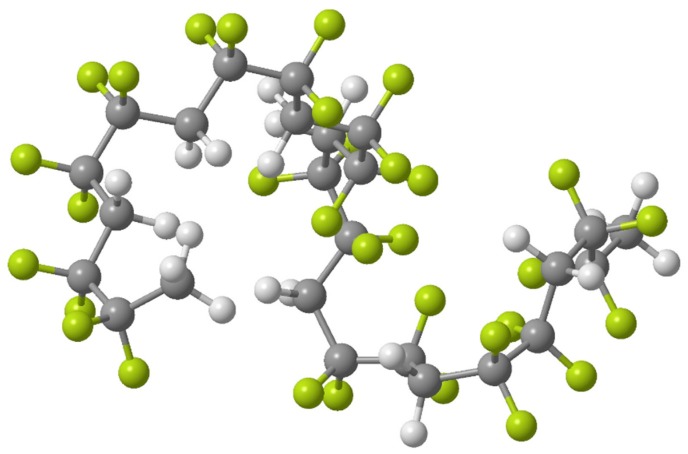
Chemical structure of the polytetrafluoroethylene (PTFE) chain.

**Figure 2 polymers-12-00306-f002:**
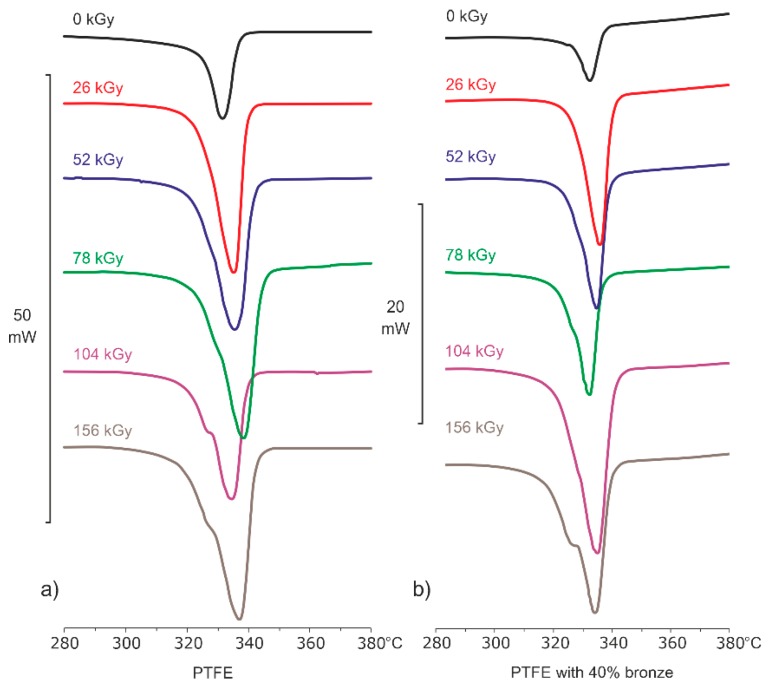
Differential scanning calorimetry (DSC) thermograms for (**a**) PTFE and (**b**) PTFE with 40% bronze in their initial states and after electron beam irradiation, registered during the heating process.

**Figure 3 polymers-12-00306-f003:**
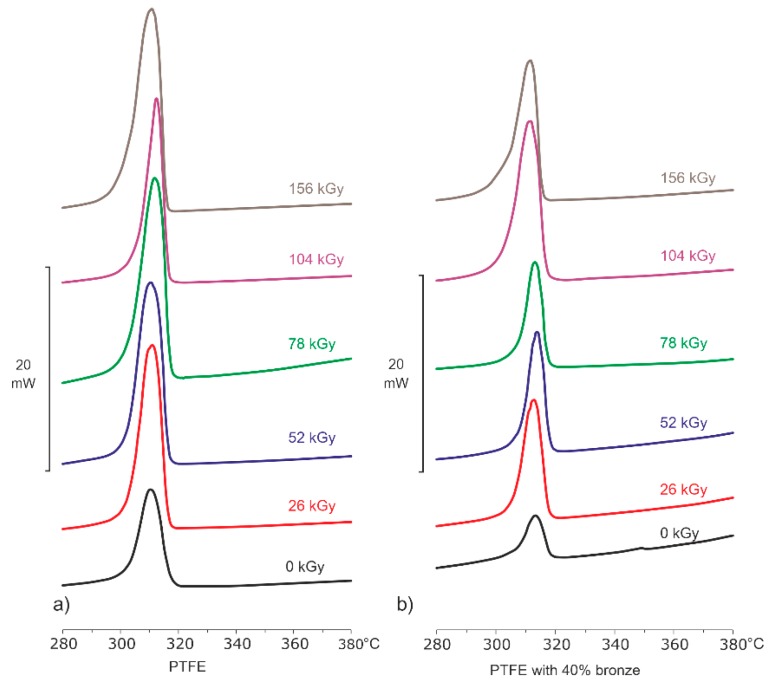
DSC thermograms for PTFE (**a**) and PTFE with 40% bronze (**b**) in their initial states and after electron beam irradiation, registered during the recrystallization process.

**Figure 4 polymers-12-00306-f004:**
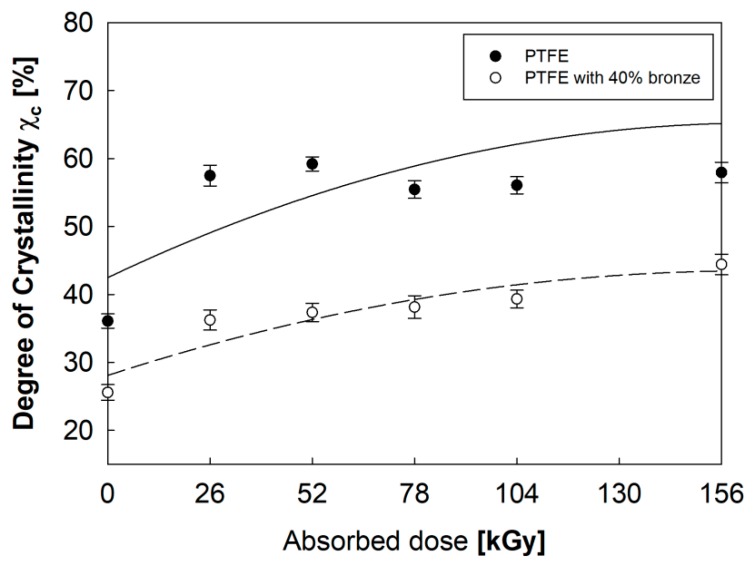
Degrees of crystallinity, *χ*_c_ of PTFE and PTFE with a 40% addition of bronze in their initial states and after electron beam irradiation.

**Figure 5 polymers-12-00306-f005:**
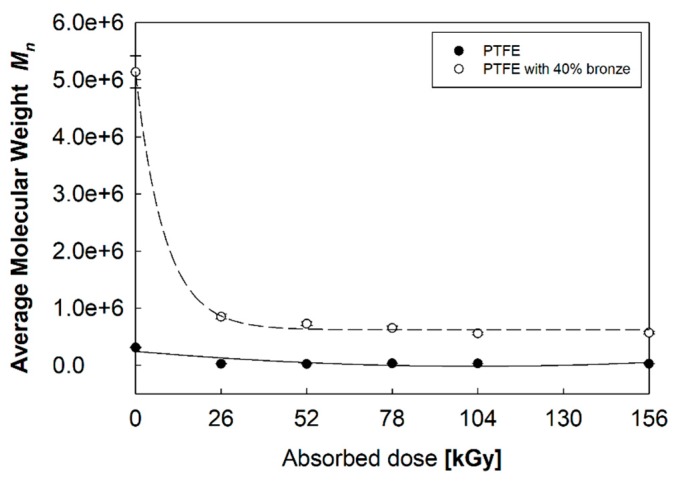
Variation of the average molecular weight *M_n_* of PTFE and PTFE with a 40% addition of bronze in their initial states and after electron beam irradiation.

**Figure 6 polymers-12-00306-f006:**
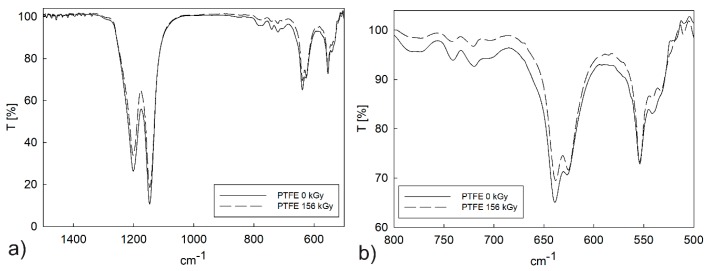
Fourier transform infrared spectroscopy (FTIR) spectra for pure PTFE in the initial state (**a**) and after a dose of 156 kGy electron beam irradiation (**b**).

**Figure 7 polymers-12-00306-f007:**
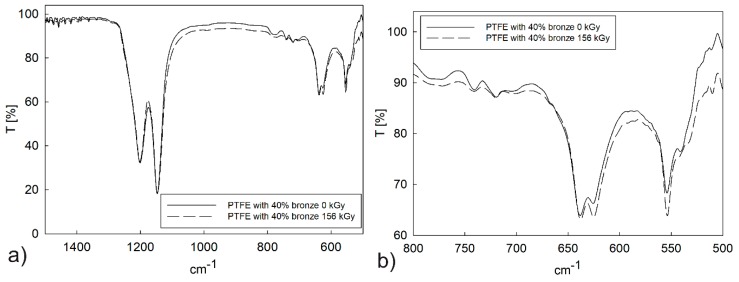
FTIR spectra for PTFE with a 40% addition of bronze in its initial state (**a**) and after electron beam irradiation with a dose of 156 kGy (**b**).

**Figure 8 polymers-12-00306-f008:**
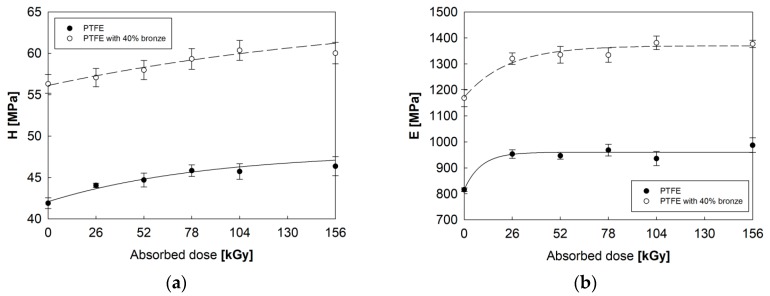
Changes in hardness H (**a**) and Young’s modulus E (**b**) for PTFE and PTFE with a 40% addition of bronze as a function of absorbed radiation dose.

**Figure 9 polymers-12-00306-f009:**
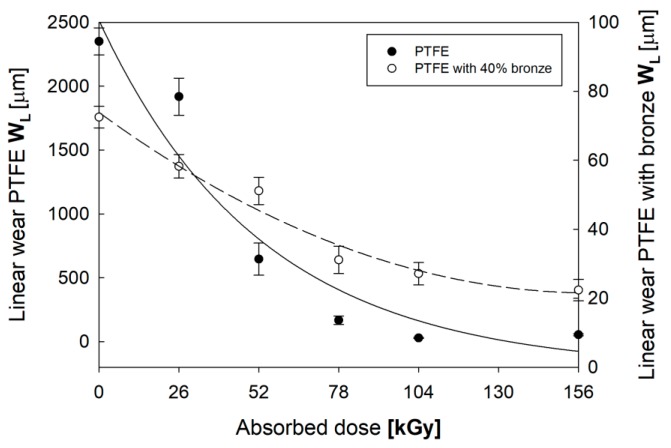
Changes in the linear wear of *W*_L_ of PTFE and PTFE with 40% bronze as a function of the absorbed radiation dose.

**Figure 10 polymers-12-00306-f010:**
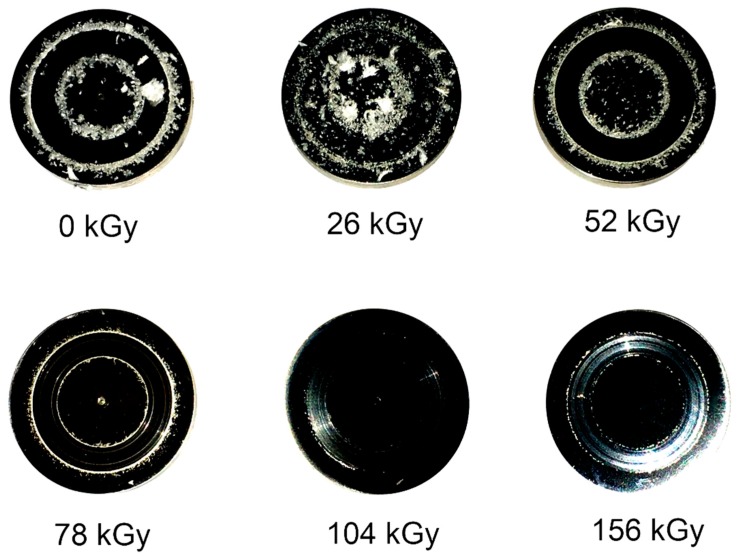
Images of 1H18N9T steel discs after tribological cooperation with pins made of PTFE in the initial state and after electron beam irradiation.

**Table 1 polymers-12-00306-t001:** Thermal properties of pure PTFE in its initial state and after electron beam irradiation.

Absorbed Dose (kGy)	*T*_m_ (°C)	Δ*H*_m_ (J/g)	*T*_c_ (°C)	Δ*H*_c_ (J/g)	*χ*_c_ (%)	*M* _n_
0	330.72	37.92	311.39	36.09	44.01	3.13 × 10^5^
26	333.85	62.95	312.37	57.48	70.10	2.83 × 10^4^
52	334.12	64.06	311.71	59.20	72.20	2.43 × 10^4^
78	337.20	67.30	313.47	55.44	67.61	3.41 × 10^4^
104	333.34	63.14	313.67	56.07	68.38	3.22 × 10^4^
156	335.57	64.66	312.21	57.93	70.65	2.72 × 10^4^

**Table 2 polymers-12-00306-t002:** Thermal properties of PTFE with 40% bronze, in its initial state and after electron beam irradiation.

Absorbed Dose (kGy)	*T*_m_ (°C)	Δ*H*_m_ (J/g)	*T*_c_ (°C)	Δ*H*_c_ (J/g)	*χ*_c_ (%)	*M* _n_
0	332.38	15.72	313.67	20.98	25.59	5.14 × 10^6^
26	335.18	39.30	313.58	29.72	36.24	8.52 × 10^5^
52	334.06	40.21	314.74	30.63	37.35	7.29 × 10^5^
78	331.42	42.04	313.68	31.28	38.15	6.54 × 10^5^
104	333.92	40.11	312.51	32.25	39.33	5.59 × 10^5^
156	333.15	42.08	312.52	32.11	39.16	5.71 × 10^5^

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
