# Peer review of "Novel Organic Material Induced by Electron Beam Irradiation for Medical Application"

_polymers, 2020, doi:10.3390/polym12020306_

Round 1
Reviewer 1 Report
This paper describes the addition of bronze to PTFE to create a composite material, and to test the effects of electron beam irradiation on this composite compared to pure PTFE. A suite of analysis is undertaken for this comparison.
Overall the paper is generally well written and frames the work well. My main question is around the comparison of the raw and composite PTFE samples, and with some of the data interpretation.
The method of sample preparation is mentioned but not discussed. The control sample of pure PTFE was prepared by extrusion, whereas the composite sample was prepared by cold pressing and sintering.
Did the preparations result in samples of a different density and so have different attenuation and penetration of the electron beam?Additionally, extrusion will necessarily result in a degree of chain alignment, which is unlikely to be present in the pressed sample to the same degree.
Was the influence this might have on the results considered? As the materials may be anisotropic, was the orientation of the material considered when the samples for wear testing were prepared?From the methods, it wasn’t clear if the FTIR analysis undertaken was undertaken using a transmittance method or ATR.
I also find the trendlines in Figure 4, Figure 8 and Figure 9 potentially misleading. For example, in Figure 4 the trendlines are both suggesting that crystallinity in decreasing with further increases in Absorbed dose. However, looking at the numerical data in Table 1 and 2 this doesn’t appear to be the case. The trend lines should be changed to better reflect what can be deduced with certainty from the data.
Line 134 could be confusing, please change – to =, so “temperature = 21 ± 1 °C”
Line 137 to 142, should this text be there?
Figure 2 – it isn’t clear which data set is which. Please add clear labels.
Line 105 – Remove the apostrophe to give “polymers”
The section from line 220-225 should be rewritten as it isn’t possible to understand that is being made here.
Author Response
Comments and Suggestions for Authors
This paper describes the addition of bronze to PTFE to create a composite material, and to test the effects of electron beam irradiation on this composite compared to pure PTFE. A suite of analysis is undertaken for this comparison.
Authors would like to thank Reviewer 1 for the time and effort in reviewing the manuscript. Our responses are below each individual comment. Any alterations to the manuscript are documented here and also included as additions in the manuscript itself (in red).
Reviewer: Overall the paper is generally well written and frames the work well. My main question is around the comparison of the raw and composite PTFE samples, and with some of the data interpretation.
Answer: We agree with the expert reviewer and the manuscript was revised follow the reviewers suggestions.
Reviewer: The method of sample preparation is mentioned but not discussed. The control sample of pure PTFE was prepared by extrusion, whereas the composite sample was prepared by cold pressing and sintering.
Did the preparations result in samples of a different density and so have different attenuation and penetration of the electron beam?
Answer: PTFE used in this study, both without additives and with a 40% bronze content, was commercially produced by Inbras in Tarnów PL, which prepares the material by both extrusion and sintering. The information provided by the company shows that in the case of PTFE without additives, the density distribution are insignificant (in the case of pressing and sintering 2.14-2.21 g / cm3, in the case of extrusion 2.13-2.22 g / cm3). When it comes to PTFE with 40% bronze content, the composite density is 3.0-3.4 g / cm3 and is related to the filler content. After irradiation with an electron beam, the effect of electron beam scattering is observed especially on filler particles, which is mainly manifested by a lower degree of crystallinity than in the case of pure PTFE (Fig. 4). The use of homogeneous ingredients and a thorough mixing process before pressing by Inbras means that the material is uniform throughout the volume. This was also confirmed by mechanical and spectrometric tests where the standard deviation of the measurements is small over the entire surface of the rod and no anisotropy is observed.
Reviewer: Additionally, extrusion will necessarily result in a degree of chain alignment, which is unlikely to be present in the pressed sample to the same degree.
Was the influence this might have on the results considered? As the materials may be anisotropic, was the orientation of the material considered when the samples for wear testing were prepared?
From the methods, it wasn’t clear if the FTIR analysis undertaken was undertaken using a transmittance method or ATR.
Answer: The text has been corrected.
Reviewer: I also find the trendlines in Figure 4, Figure 8 and Figure 9 potentially misleading. For example, in Figure 4 the trendlines are both suggesting that crystallinity in decreasing with further increases in Absorbed dose. However, looking at the numerical data in Table 1 and 2 this doesn’t appear to be the case. The trend lines should be changed to better reflect what can be deduced with certainty from the data.
Answer: The trends lines in mentioned figures (Fig 4, 8, 9) were corrected.
Reviewer: Line 134 could be confusing, please change – to =, so “temperature = 21 ± 1 °C”
Answer: The authors agree with the reviewer’s comment and the text has been corrected.
Reviewer: Line 137 to 142, should this text be there?
Answer: The authors agree with the reviewer’s comment and the text was removed.
Reviewer: Figure 2 – it isn’t clear which data set is which. Please add clear labels.
Answer: The labels was added.
Reviewer: Line 105 – Remove the apostrophe to give “polymers”
Answer: The authors agree with the reviewer’s comment and the apostrophe was removed.
Reviewer: The section from line 220-225 should be rewritten as it isn’t possible to understand that is being made here.
Answer: The authors agree with the reviewer’s comment and the section has been rewritten.

Reviewer 2 Report
The manuscript studies polytetrafluoroethylene with 40% bronze by experimental methods. The material is treated with irradiation of electron beam to modify its properties. Experimental analysis of DSC, FTIR, microindentation and tribological tests. It is recommended that manuscript be accepted with minor revision. Review comments are listed as follows.
1. Repeated sentences in Lines 40-45.
2. On Lines 80 and 228, the 'grain size of 0-32' statement is problematic. How can grain size to be zero?
3. On Line 85, please explain the 'its multiple i=1-6' statement.
4. On Line 121, it should read 'Oliver-Pharr'.
5. Although the manuscript is of experimental type, it would be helpful to provide some theoretical estimates on the physical properties of such polymer composite material.
6. Irradiation modified the surface properties of the material, which may cause brittleness and detrimental effects on long-term properties. Please provide surface images of the samples after microindentation tests to make sure no cracks occur around the indents due to the effects of brittleness. Please provide discussions on possible detrimental effects on long-term properties of such irradiation-treated samples.
7. Please provide how uniform the bronze particles are distributed in PTFE samples.
Author Response
Comments and Suggestions for Authors
The manuscript studies polytetrafluoroethylene with 40% bronze by experimental methods. The material is treated with irradiation of electron beam to modify its properties. Experimental analysis of DSC, FTIR, microindentation and tribological tests. It is recommended that manuscript be accepted with minor revision. Review comments are listed as follows.
The authors would like to thank Reviewer 2 for the time and effort in reviewing the manuscript. Our responses are below each individual comment. Any alterations to the manuscript are documented here and also included as additions in the manuscript itself (in red).
Reviewer: Repeated sentences in Lines 40-45.
Answer: The repeated sentence was removed.
Reviewer: On Lines 80 and 228, the 'grain size of 0-32' statement is problematic. How can grain size to be zero?
Answer: The text has been corrected.
Reviewer: On Line 85, please explain the 'its multiple i=1-6' statement.
Answer: The statement may have been incomprehensible, this piece of text has been removed. The idea was that the samples were irradiated from 26 - 156 kGy, with 26 kGy doses or its multiplicities, i.e. in the case of 26 kGy once and 156 kGy 6 times.
Reviewer: On Line 121, it should read 'Oliver-Pharr'.
Answer: The text has been corrected.
Reviewer: Although the manuscript is of experimental type, it would be helpful to provide some theoretical estimates on the physical properties of such polymer composite material.
Answer: Theoretical simulations were not carried out, authors concluded that the article will gain more on the comparison of practical test results for both materials.
Reviewer: Irradiation modified the surface properties of the material, which may cause brittleness and detrimental effects on long-term properties. Please provide surface images of the samples after microindentation tests to make sure no cracks occur around the indents due to the effects of brittleness. Please provide discussions on possible detrimental effects on long-term properties of such irradiation-treated samples.
Answer: Micromechanical tests were carried out at 1N load, 30s load rise time, 10s holding time under full load and 30s unloading time. The load-unload curve was recorded in real time for each measurement. In the case of PTFE without additives and with 40% bronze content, there were no cracks, as evidenced by the course of indentation curves (Figure a and b), on which the characteristic steps occurring during cracking were not recorded. For example, a load-unload curve for brittle material c) is shown. Nevertheless, a partial deterioration of mechanical properties and an increase in tribological wear were observed for the 156 kGy dose, which may indicate material degradation occurring at high doses of electron beam irradiation.
The authors decided not to put these test results so that the article was not too long. However, maybe these tests should be added as supplementary data to dispel the reader's doubts.
Reviewer: Please provide how uniform the bronze particles are distributed in PTFE samples.
Answer: PTFE used in this study, both without additives and with a 40% bronze content, was commercially produced by Inbras in Tarnów PL, which prepares the material by both extrusion and sintering. The information provided by the company shows that in the case of PTFE without additives, the density distribution are insignificant (in the case of pressing and sintering 2.14-2.21 g / cm3, in the case of extrusion 2.13-2.22 g / cm3). When it comes to PTFE with 40% bronze content, the composite density is 3.0-3.4 g / cm3 and is related to the filler content. After irradiation with an electron beam, the effect of electron beam scattering is observed especially on filler particles, which is mainly manifested by a lower degree of crystallinity than in the case of pure PTFE (Fig. 4). The use of homogeneous ingredients and a thorough mixing process before pressing by Inbras means that the material is uniform throughout the volume. This was also confirmed by mechanical and spectrometric tests where the standard deviation of the measurements is small over the entire surface of the rod and no anisotropy is observed.

Round 2
Reviewer 1 Report
I have reviewed the responses and changes from the authors and I am happy that the paper can be published
Author Response
Thank you to the reviewer for your time and positive assessment of our work.